# The Material Matters: Sorption/Desorption Study of Selected Estrogens on Common Tubing or Sampling Materials Used in Water Sampling, Handling, Analysis or Treatment Technologies

Klára Odehnalová , Petra Přibilová, Štěpán Zezulka and Blahoslav Maršálek *

Institute of Botany, Czech Academy of Sciences, Lidická 25/27, 60200 Brno, Czech Republic
* Correspondence: marsalek@ibot.cas.cz

**Abstract:** Plastic/rubber materials used as fasteners in equipment for analyzing or removing organic pollutants in water treatment technologies form an essential part of the device. Micropollutants in water are typically present at very low concentrations (ng/L to µg/L). Therefore, when designing, for example, units for advanced oxidation processes (AOPs) or planning sample handling, it is necessary to assess whether the material is compatible with the usually hydrophobic nature of the pollutants. As a model example, the possible interactions of estrogens, namely, estrone (E1), 17β-estradiol (E2), estriol (E3) and 17α-ethinylestradiol (EE2) with six commonly used plastic and rubber materials were investigated at environmentally relevant concentrations (100–500 ng/L). In the first phase, we proved that polyvinyl chloride (PVC), polytetrafluoroethylene (PTFE), polyvinylidene fluoride (PVDF) and ethylene propylene diene monomer (EPDM) materials adsorbed only negligible amounts of estrogens, while significant amounts of E1, E2 and EE2 were adsorbed onto Tygon S3™ material. Another unsuitable material was styrene butadiene rubber (SBR), sorbing a considerable quantity of estrone. A detailed test of EPDM at higher concentrations (300 and 500 ng/L) and prolonged soaking time showed significant sorption of EE2 after 12 h of soaking in both deionized and tap water matrices. Thus, EPDM, PTFE and PVDF are suitable materials for sample handling or producing devices for AOP treatment due to their chemical inertness and mechanical flexibility. The results suggest that plastic materials that come into contact with contaminated water must be carefully selected, especially when working at environmentally relevant concentrations.

**Keywords:** sorption; desorption; plastics; estrogens; hormones; environmentally relevant concentrations; tubing materials; sampling materials





## 1. Introduction

Plastic and rubber materials or materials with a polymeric coating [1] are currently used in various applications, including laboratory or technological equipment. Except for disposable items (pipette tips, vials, containers, gloves, etc.), many of these materials are designed for repeated use, such as instrument hoses. Tubing materials are desired to be durable, inert and resistant to mechanical and chemical irritation, especially in applications such as advanced oxidation processes (AOP), where specific oxidative and irritating conditions can occur [2–4]. In addition to chemical resistance, resistance to fluctuating and abrasive process conditions, thermal conductivity, fabrication capabilities and design play significant roles in the decision-making process. Nevertheless, when used in contact with hydrophobic compounds, their polymeric nature poses a risk of contamination. Unwanted sorption (adsorption/absorption) and desorption events of trace amounts of organic pollutants can cause severe discrepancies during sample handling and analyses [5,6], especially when only trace and environmentally relevant quantities are present. Among factors that contribute to sorption may be the ability of a micropollutant to interact specifically with polymer functional groups, Gibbs free energy of the localized environment, and other

factors. Moreover, the sorption could be enhanced by the surface roughness and microporosity of the polymer material. Due to the predominantly hydrophobic nature of many organic pollutants, polymeric materials are expected to interact with them through weak hydrophobic interactions, van der Waals forces or noncovalent interactions [5,7,8]. Between them, natural estrogenic hormones, such as estrone (E1), 17β-estradiol (E2), estriol (E3) and synthetic contraceptive 17α-ethinylestradiol (EE2) are emerging micro-pollutants established in the Watch List of Substances for European Union-wide monitoring reported in the Decision 2015/495/EU of 20 March 2015 [9] and US EPA [10]. Due to their hydrophobicity values expressed as log $K_{ow}$ (see Table S1) and the high-risk coefficients, estrogens represent relevant model compounds [11,12]. They are released into the environment via wastewater-treated effluents as they are poorly removed with conventional wastewater treatment plant (WWTP) processes [13–16]. Various studies have reported their occurrence in aqueous and solid environmental matrices and tap water [17–21]. Although their concentrations in the environment were found in the ng/L range, they show high estrogenic potency. The presence and persistence of estrogens in the environment raise concerns regarding the potential adverse effects on the sexual and reproductive systems of wild animals, fish and humans [22–26]. The ability of estrogens to sorb onto the surfaces of various materials is used for their removal from the environment [27–31]. On the other hand, the sorption and following desorption of such chemicals can be a source of problems. For example, due to organic carbon as a sorbent (e.g., humic acids), they can reach deeper layers of soil (possessing divergent conditions), where they can be desorbed and thus contaminate groundwater [26,32]. Similarly, plastic waste [33], especially microplastics [34,35] as an important part of current environmental pollution, can serve as a source and vector for various chemicals in aquatic and terrestrial environments. The sorption of estrogens can also be a source of inaccuracy in analysis, especially when their concentrations are very low, as is usually the case in environmental samples. If estrogens sorb to the sample handling material or materials used for sample filtration, the results may be significantly underestimated or false negative [5,25]. In addition, if they sorb to coupling material in AOP units, better degradation efficiencies may be falsely attributed to sorption.

Given the above, estrogens were selected as model compounds for sorption/desorption studies on materials commonly used for sample handling or piping in laboratory devices in this study. The main hypothesis of this study was that sorption/desorption processes might cause inaccuracies in the analysis (or removal) of estrogenic compounds in water samples, especially when present at low, environmentally relevant concentrations. The extent of sorption and desorption of three natural estrogens (E1, E2 and E3) and one synthetic (EE2) to and from six different polymeric materials was investigated using LC-MS/MS techniques.

## 2. Materials and Methods

### 2.1. Chemicals and Materials

Standard compounds of estrone (E1; ≥99%), 17β-estradiol (E2; ≥98%), estriol (E3; ≥98%) and 17α-ethinylestradiol (EE2; ≥98%; for an overview of physicochemical properties see Table S1 in Supplementary Materials), as well as all solvents and reagents, were purchased from Sigma-Aldrich (St. Louis, MO, USA). Internal standard 17β-estradiol-d4 (E2-d4) and 17α-ethinylestradiol-d4 (EE2-d4) were purchased from C/D/N Isotopes Inc. (Pointe-Claire, Quebec, Canada). All standards were prepared in HPLC-grade methanol. Methanol, acetonitrile and acetone (HPLC grade), formic acid, hydrochloric acid, dansylchloride and sodium bicarbonate were used. All tested materials/tubes were purchased from Gumex, spol. s r.o. (Brno, Czech Republic). Material characteristics and suggested applications are shown in Tables 1 and S2.

**Table 1.** Tested materials for estrogen sorption/desorption.

| Material | Characteristics [a] | Applications |
|---|---|---|
| Tygon S3™ E-3603 | non-DEHP PVC, non-toxic, non-contaminating, chemical resistance, non-oxidizing, glassy–smooth inner bore | analytical instruments, general laboratory, peristaltic and vacuum pumps, biopharmaceutical, incubators, desiccators, gas lines, food, beverages |
| Polyvinyl chloride (PVC) | resistant to weathering, chemical rotting, corrosion, shock and abrasion; self-extinguishing | water, gas, sewage, industrial process, irrigation, medical devices, blood storage bags |
| Teflon (PTFE) | chemical inertness, high resistance to ageing, temperature resistance | conveying pressures and temperatures, fluids, corrosive fluids, steam, push–pull cables, seals, gaskets |
| Kynar® (PVDF) | outstanding resistance to UV exposure, tremendous chemical resistance to a wide range of aggressive chemicals, resistance to chemical products, soluble in aprotic solvents | contact surface for the production, storage and transfer of corrosive fluids, used in mechanical components, fabricated vessels, tanks, pumps, valves, filters, heat exchangers, tower packing, piping systems, seals, gaskets |
| Ethylene propylene diene monomer (EPDM) | resistant to ozone and weather, abrasion resistant, UV resistant, electrically conductive, cloth impression | suitable for use as a discharge hose in the chemical industry and raw materials extraction |
| Styrene butadiene rubber (SBR) | good resilience and tensile strength, outstanding resistance to abrasion and fatigue; water, organic acid, ketone, chemical, alcohol, and aldehyde resistance; low resistance to ozone | industrial applications, adhesives, rubber/mechanical goods, car tires |

[a] Selected physical characteristics are summarized in Table S2.

### 2.2. Short-Term Material Exposure

In the first phase of the study, the sorption experiments were carried out with six materials: styrene butadiene rubber (SBR), ethylene propylene diene monomer (EPDM), polytetrafluoroethylene (PTFE), Tygon S3™ E-3603, KYNAR® (polyvinylidene fluoride, PVDF) and polyvinyl chloride (PVC). A piece of each material with a surface area of 1000 cm$^2$ was individually immersed in a bath (2 L) with deionized water containing a mixture of estrogens with a concentration of 100 ng/L each. The content was continuously stirred with a magnetic stirrer at 400 rpm at 21 °C. After 30 min of incubation, the solution was sampled (50 mL) for analysis of estrogen content. For desorption assessment, the estrogen solution was replaced with clean deionized water and stirred again under the same conditions. Samples (50 mL) for desorbed estrogen analyses were taken out after 30, 60 and 90 min.

### 2.3. Long-Term EPDM Exposure

For further testing, EPDM was chosen, and the sorption period was expanded to 1, 3, 5 and 12 h using estrogen concentrations of 300 and 500 ng/L. Soaking with estrogens was carried out in deionized water (pH = 5.8 ± 0.1, κ < 0.05 µS/cm) as well as in tap water (pH = 7.1 ± 0.1; κ = 417 ± 2.0 µS/cm) matrix. After 12 h exposure, the possible estrogen desorption after 30, 60 and 90 min back into deionized water was tested.

### 2.4. LC-MS/MS Analysis

2.4.1. Solid Phase Extraction and Derivatization

Extraction of the estrogen from samples was performed using SPE Oasis® HLB (500 mg, 6 mL) cartridges (Waters, Milford, MA, USA) according to Sadilek et al. [36] with minor modifications. The cartridges were pre-conditioned using 5 mL of methanol followed by 5 mL of Milli-Q water. Acidified samples (pH = 3 ± 0.2) were loaded onto

the cartridges and extracted under vacuum at a flow rate of approximately 5 mL/min. Afterwards, the sample holders were rinsed with 5 mL of distilled water and dried under vacuum suction for 20 min. Once the extraction was completed, analytes were eluted with 8 mL of methanol, and a mixture of internal standards (20 μL, 25 ng/mL) was added and dried under a gentle nitrogen stream at 40 °C.

The dried extract was reconstituted in 20 μL of acetone, and then 50 μL sodium bicarbonate buffer (100 mM, pH = 10.5) and 50 μL dansylchloride (1 mg/mL in acetone) was added. The reaction mixture was vigorously shaken for 1 min and tempered at 60 °C for 5 min. After cooling to the laboratory temperature, the mixture was allowed to dry under a gentle nitrogen stream at 55 °C. Samples were then reconstituted in 1 mL of methanol:water mixture (40:60; *v/v*) and filtered through a syringe filter (nylon; 0.45 μm) before analysis.

### 2.4.2. Instrumental Analysis

An Agilent 1260 Infinity HPLC system combined with an Agilent 6460 TripleQuad mass spectrometer (Agilent Technologies, Santa Clara, CA, USA) equipped with an electrospray ionization interface (ESI) was used to quantify estrogens. Separation was achieved using Poroshell 120 EC-C18 (2.1 × 100 mm; 2.7 μm) fitted with a security guard column of the same packing material (Agilent Technologies, Santa Clara, CA, USA). Mobile phases A and B consisted of formic acid (7 mM) and acetonitrile, respectively. The solvent gradient program concerning mobile phase B was as follows: 0 min 50%; 0–10 min 100%; 10–11 min 100%; and 11–11.1 min 50%, followed by equilibration for 5 min. The flow rate was 350 μL/min, and the injection volume was 10 μL. Mass data were acquired using MassHunter Workstation software (Agilent Technologies, Santa Clara, CA, USA) and multiple reaction monitoring (MRM) in the positive mode. The detector settings were capillary voltage 3500 V, nozzle voltage 2000 V, gas temperature ($N_2$) 200 °C, gas flow 10 mL/min, nebulizer 50 psi, sheath gas temperature ($N_2$) 350 °C and sheath gas flow 10 mL/min. The precursor and product ions, collision energy and fragmentor voltage used for MRM are summarized in Table 2.

**Table 2.** Multiple reaction monitoring (MRM) transitions and operating MS/MS parameters of estrogen dansyl derivatives.

| Compound | Precursor Ion (m/z) | Quantitation/Qualification Ion (m/z) | Collision Energy (V) | Fragmentor Voltage (V) |
|---|---|---|---|---|
| E1 | 504 | 171/156 | 140 | 38 |
| E2 | 506 | 171/156 | 140 | 42 |
| EE2 | 530 | 171/156 | 140 | 40 |
| E3 | 522 | 171/156 | 140 | 42 |
| E2-d4 | 510 | 171/156 | 140 | 42 |
| EE2-d4 | 534 | 171/156 | 140 | 40 |

### 2.5. Data Analysis and Quality Control

All sorption/desorption experiments were performed in duplicate. Confirmation of estrogens in samples was based on the MRM ion transitions and comparing the retention time of each peak to the corresponding standard. The method's performance was evaluated by considering both water matrices' response linearity, method detection and quantification limits, recovery (Table S3) and repeatability (intra- and inter-day variations).

### 2.6. Statistical Evaluation

Statistical analyses were performed using GraphPad Prism software. A one-way ANOVA on ranks followed by a Dunn test was used to determine differences between the control (initial concentration) and the treated samples (concentration of estrogens in solution after incubation period). The differences between individual means were indicated using Mann–Whitney test. A probability level of $p < 0.05$ was considered statistically significant. All data are represented as mean ± standard deviation of relative concentrations (related

to initial concentration). Relative adsorbed or desorbed concentration was calculated as follows:

$$\text{Sorption [\%]} = (1 - C/C_0) * 100 \tag{1}$$

$$\text{Desorption [\%]} = (C/C_0) * 100 \tag{2}$$

where C is the actual concentration in the solution, and $C_0$ denotes the initial concentration in ng/L.

## 3. Results and Discussion

### 3.1. LC-MS/MS Analysis

The quantification of estrogens was performed after their preconcentration on solid phase and dansylation using the LC-MS/MS method. An example of a chromatogram is shown in Figure 1. Matrix-matched calibration curves (5 points ranging from 5 to 600 ng/L) with standards of estrogens were used for quantification with the $r^2$ of at least 0.99 for all analytes each day. Limits of detection (MDL) and quantification (MQL) were calculated based on a signal-to-noise ratio of 3.3 and 10, respectively, and their values are shown in Table 3. The extraction recoveries for each matrix and concentration level are listed in Table 4.

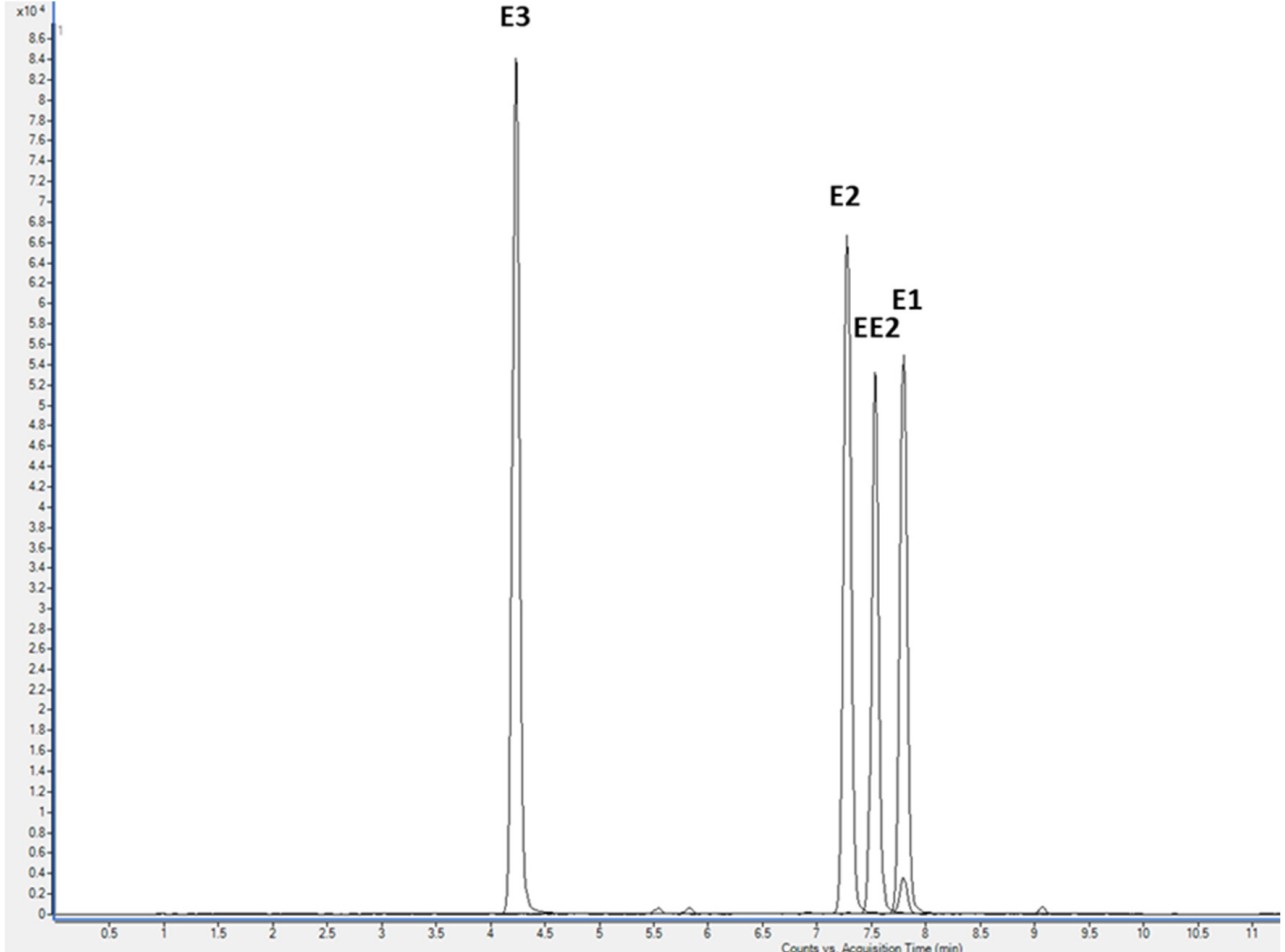

**Figure 1.** Chromatogram of estrogen dansyl derivatives (300 ng/L in deionized water).

**Table 3.** Method validation parameters.

| Compound | MDL (ng/L) | MQL (ng/L) | CV [a] (%) Inter-Day | CV [a] (%) Intra-Day | CV [b] (%) Inter-Day | CV [b] (%) Intra-Day | CV [c] (%) Inter-Day | CV [c] (%) Intra-Day |
|---|---|---|---|---|---|---|---|---|
| E1 | 0.2 | 0.5 | 7.54 | 10.41 | 12.22 | 11.59 | 11.18 | 9.97 |
| E2 | 0.2 | 0.5 | 4.47 | 7.33 | 13.03 | 13.08 | 5.29 | 4.51 |
| EE2 | 0.1 | 0.4 | 3.69 | 7.65 | 8.88 | 8.76 | 5.62 | 4.50 |
| E3 | 0.2 | 0.6 | 6.43 | 3.84 | 13.25 | 13.28 | 13.94 | 15.09 |

[a] 100 ng/L—deionized water; [b] 300 ng/L—tap water; [c] 500 ng/L—tap water; CV = covariance; MDL = limit of detection; MQL = limit of quantification.

**Table 4.** Recoveries (%) of estrogens at individual concentration levels and matrices.

| Concentration | E1 | E2 | EE2 | E3 |
|---|---|---|---|---|
| 100 ng/L [a] | 97.06 | 100.35 | 99.03 | 123.84 |
| 300 ng/L [a] | 73.15 | 83.34 | 88.52 | 94.11 |
| 300 ng/L [b] | 88.47 | 92.43 | 92.91 | 95.17 |
| 500 ng/L [a] | 114.33 | 104.61 | 111.48 | 83.62 |
| 500 ng/L [b] | 96.80 | 107.75 | 103.07 | 113.09 |

[a] Deionized water; [b] tap water.

### 3.2. Short-Term Material Exposure

For the first screening of estrogen sorption, six materials generally recommended for chemical handling were evaluated after a 30-min soak in a solution consisting of deionized water spiked by 100 ng/L of each estrogen. As using contaminated equipment can cause severe inaccuracies in the processing of samples, we also tested whether the estrogens were desorbed back from the material surface into the deionized water. The results of the sorption experiment are shown in Figure 2a, and the desorption experiments are shown in Figure 2b.

As shown in Figure 2a, significant changes in concentrations were observed in the case of estrone having the highest $logK_{0W}$ value (Table S1) and especially on SBR, where almost 50% of the initial concentration in solution was sorbed. Moreover, only 2% of the E1 initial concentration was desorbed back to the deionized water after 90 min, suggesting quite strong, probably hydrophobic, interactions. All estrogens in our study contain a phenolic hydroxyl group. Thus, they are strong OH donors and acceptors but weak $\pi$ acceptors. Estrone, however, also includes a keto group. Therefore, another possible but less likely explanation could be the formation of an OH/$\pi$ hydrogen bond between the enol-form of estrone and the unsaturated group of SBR. However, the enol form is generally less stable in aqueous media.

Significant sorption (more than 20%) of all estrogens was observed on Tygon S3™, a DEHP-free PVC. This material contains a bio-based plasticizer. Generally, plasticizers increase the space between the polymer chains, thus increasing the material's permeability [37]. As plasticizers are small molecules, they can leach out of plastics. In the same way that deliberately added small molecules could drain out, the plastic can absorb small molecules from the environment. Following the results of desorption from this material, nearly half of the previously sorbed estrogens were desorbed after 90 min soaking with deionized water. This suggests that the estrogens did not interact with the material as we observed for estrone and SBR. They probably interact with common but weaker (2 kJ/mole) dispersion forces or could just reversibly permeate into it (be absorbed). The estrogens were released continuously from the surface throughout the time frame with maximal concentration (approx. 5% of initial concentration) during the first 30 min. After 90 min, the desorption of E1, E2 and EE2 reached nearly 10% of their initial concentration (Figure 2b). It should also be noted that Tygon S3™ has the highest water absorption values (Table S2). Due to the high sorption and subsequent continuous desorption of estrogens into the pure medium, Tygon S3™ was found to be an unsuitable material. A similar situation in

desorption was observed in the case of PVC. However, the estrogen sorption on PVC was insignificant ($p < 0.05$).

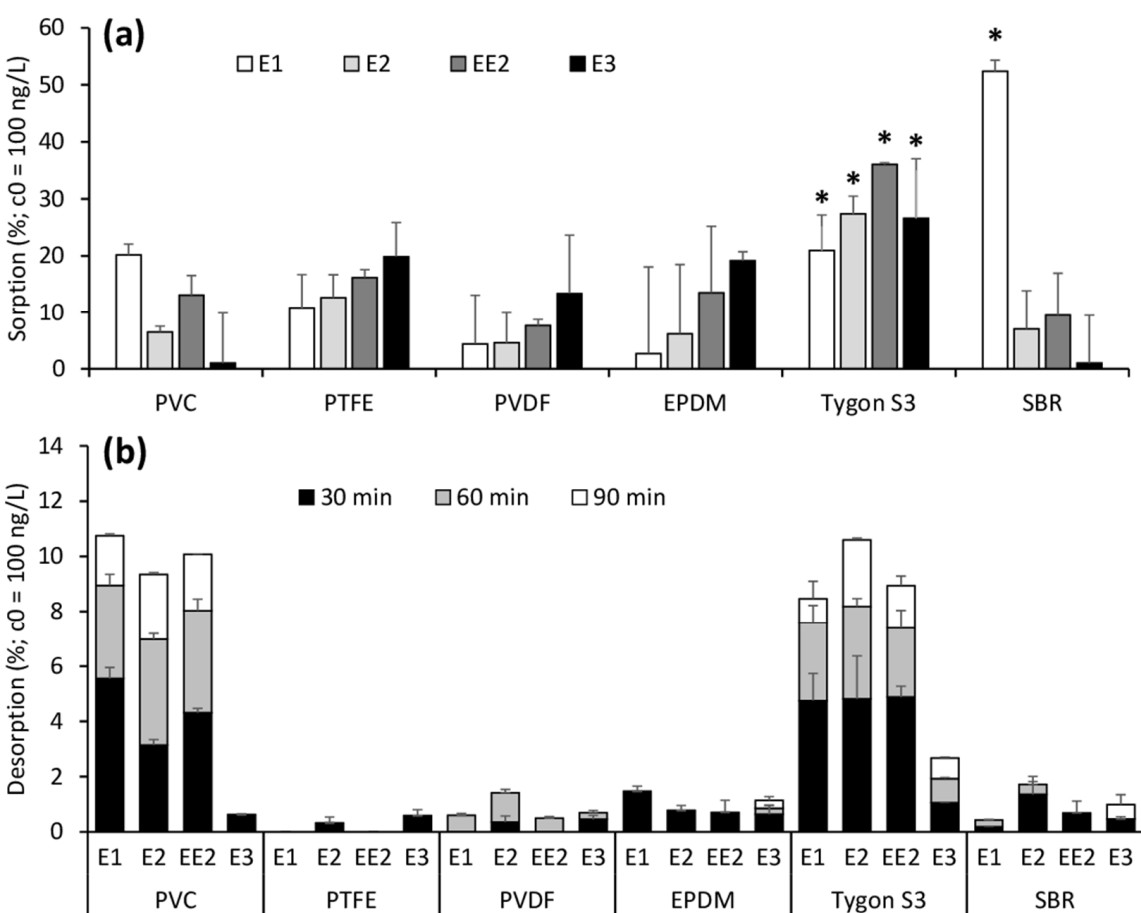

**Figure 2.** Screening of the relative estrogen sorption (**a**) after 30 min of incubation of six polymeric materials in the deionized water with estrogens (E1, E2, EE2 and E3) and the relative cumulative estrogen desorption (**b**) after 30, 60 and 90 min incubation of the contaminated materials in clean deionized water. The initial concentration (c0) of all estrogens was 100 ng/L. Data represent the mean ± standard deviation of relative concentrations (related to initial concentration). Asterisks mark significant differences detected using ANOVA and Dunn test at $p < 0.05$.

The extent of estrogen sorption on the other materials was found to be similarly low (less than 20%). The overall desorption of estrogens from these materials was lower than 2% of the initial concentration, even after 90 min soaking in clean deionized water. Materials such as PVC, PVDF and PTFE cut into the columns (13 × 100 mm) have been tested by other authors [5] and found suitable for estrogen handling. These authors found that after 24h contact with PTFE and PVC, only minimal amounts of E2, EE2 and E1 were adsorbed. They also observed that some portion of E1, E2 and EE2 was adsorbed on PTFE (0.2 μm) and PVDF (0.45 μm) membrane filters, but the differences were lower than 20%, thus insignificant ($p < 0.05$). In contrast, Han and colleagues [27] found unexpectedly significant adsorption of estrone onto PTFE membrane filters. In two different experiments, this team demonstrated that physical sorption is involved in the case of estrone. First, they observed that sorption increases with the pore size of the PTFE membrane (80 and 42% for 0.1 and 0.45 μm pore size, respectively), which is related to the sorption surface. Then, they also supported this theory with an experiment where increasing the amount of filtered solution led to saturation of the studied material, and estrogen was no longer adsorbed onto the membrane. Although both teams worked with similar concentrations (0.5 and 0.4 mg/L, respectively) and filters from the same manufacturer, their results were

opposed. Additionally, working with membranes or small areas of tested materials in combination with such high estrogen concentrations does not reflect the conditions in environmental samples.

All tested plastic materials are commonly used in laboratories, the chemical industry or food production, including, e.g., water quality control processes. Sorption of estrogens to them can cause potential losses and thus underestimate real estrogen concentrations in the analysed samples. On the other hand, the desorption into the samples can elevate at least the background level and cause discrepancies during analyses, especially in samples with concentrations close to the limits of detection/quantification.

*3.3. Long-Term Material Exposure*

Given the results of the first part of the study and the fact that other authors already proved PVC, PTFE and PVDF for estrogen sorption, albeit they used much higher concentrations, we chose EPDM material for a more detailed evaluation. EPDM is a non-crystalline thermoplastic with similar polarity to perfluorinated polymers. Its most notable advantages are UV and ozone resistance, elasticity with 600% elongation and tensile strength (see Table S2), making it a suitable material for constructing, for example, AOP units. In this phase of the study, sorption at higher estrogen concentrations (300 and 500 ng/L) in two different matrices (deionized water and tap water) and a prolonged exposure period (1–12 h) was tested. The results of the sorption experiments are graphically illustrated in Figure 3.

As can be seen in Figure 3a,b, the only significant sorption was observed for EE2 after 12 h at both concentration levels and water matrices. The sorption of E1 and E2 reached up to 20% of the initial concentration, and the sorption of E3 was negligible in all experiments. The presence of ions in solution can also affect the sorption behavior of plastics by affecting the solubility of organic compounds in water. In general, increasing ionic strength reduces the solubility of non-polar and weakly polar organic contaminants in water, known as the salting-out effect [38]. Thus, the higher salt level increases the availability of certain hydrophobic contaminants for adsorption onto plastics. This effect was observed at a higher tested concentration (500 ng/L), whereas in tap water ($\kappa = 417 \pm 2.0$ μS/cm), the adsorbed amount of EE2 was two times higher than in deionized water.

Subsequently, after 12 h sorption experiments, desorption into the deionized water was evaluated. Estrogen cumulative concentrations desorbed from the EPDM surface are shown in Figure 3c. Although the sorbed amount of EE2 was at least 20%, the desorbed amount of all estrogens did not exceed 10% of the initial concentration at both concentration levels (300 and 500 ng/L) and both water matrices. EE2 was not leached into the deionized water, most probably due to the hydrophobic interactions. Although the highest log $K_{OW}$ value has estrone, higher EE2 adsorption was found in both matrices on EPDM. The dissociation constants (*pKa*) of selected estrogens have values around 10.3 (Table S1), indicating that in both matrices (pH of 5.8 and 7.1 for deionized water and tap water, respectively), they stay uncharged. This way, the charge interaction was minimal and not responsible for significant sorption. Another property that may affect interaction with polymers is proton donor and acceptor characteristics, in particular, the ability to form hydrogen bonds, which has been attributed to play a predominant factor in the transport of estrogens in biological systems [27,39].

The EPDM material was evaluated as the most proper, mainly regarding chemical properties. In addition to the fact that this material showed low sorption of estrogens, its mechanical properties are equally important, as these materials can be bent with a smaller radius than PTFE or PVDF, which have similar polarity. The flexibility of EPDM material can significantly save costs and reduce material consumption while eliminating the adverse effects of possible, albeit minimal, undesirable estrogen sorption.

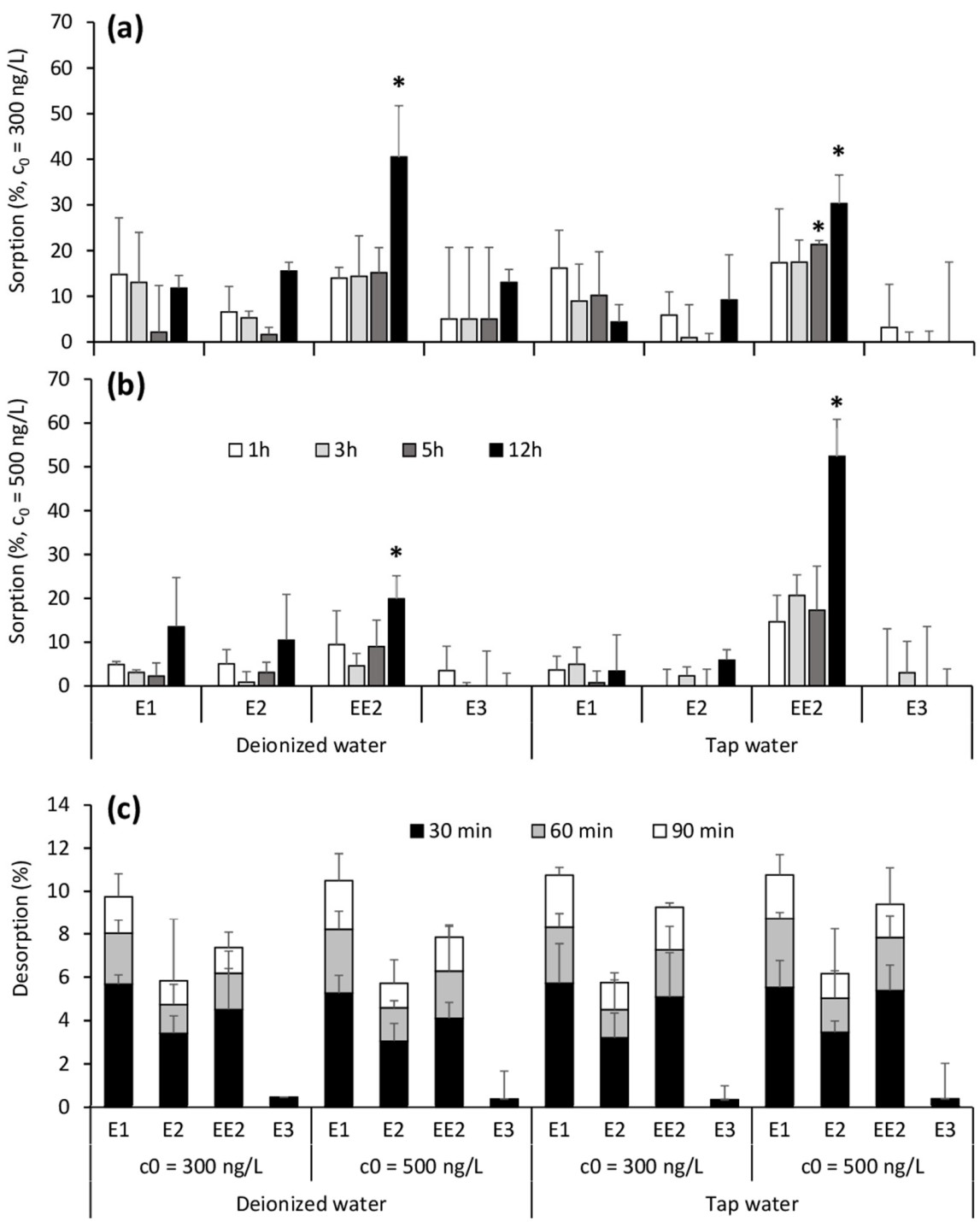

**Figure 3.** Relative sorption (**a**,**b**) after 1, 3, 5 and 12h of incubation of ethylene propylene diene monomer (EPDM) in the deionized or tap water with estrogens added (initials concentrations of 300 and 500 ng/L) and the relative desorption (**c**) after 30, 60 and 90 min incubation in clean deionized or tap water. Asterisks mark significant differences detected using ANOVA and Dunn test at $p < 0.05$.

## 4. Conclusions

The present study confirmed that plastic materials differ in their ability to sorb and desorb estrogenic compounds. After short-term exposure to low estrogen concentrations (100–500 ng/L), reflecting the real situation in environmental samples, materials such as PVDF, PTFE and EPDM exhibited only low absorption and desorption capability to E1, E2, EE2 and E3 estrogens. On the other hand, materials such as PVC, especially Tygon S3™

E-3603, absorbed and desorbed substantial amounts of estrogens in short periods (up to 30 min) and are, therefore, inappropriate for handling samples contaminated with trace amounts of estrogens. SBR material absorbed a considerable amount of E1 but without significant desorption. EPDM was proven to be appropriate for manipulation with this group of estrogens and has better mblechanical flexibility than other hard polymers such as PTFE or PVDF, which are appropriate for sample storing, handling or transporting.

**Supplementary Materials:** The following supporting information can be downloaded at: https://www.mdpi.com/article/10.3390/app13053328/s1, Table S1: Physico-chemical properties of selected estrogens; Table S2: Selected physical characteristics of tested materials; Table S3: Method validation parameters.

**Author Contributions:** Conceptualization, B.M. and K.O.; methodology, K.O.; validation, K.O. and P.P.; formal analysis, Š.Z.; investigation, K.O. and P.P.; data curation, Š.Z; writing—original draft preparation, K.O., Š.Z. and P.P.; writing—review and editing, B.M..; visualization, Š.Z.; supervision, B.M. All authors have read and agreed to the published version of the manuscript.

**Funding:** This research was funded by the Czech Science Foundation, project No. 19-10660S.

**Institutional Review Board Statement:** Not applicable.

**Informed Consent Statement:** Not applicable.

**Data Availability Statement:** Not applicable.

**Conflicts of Interest:** The authors declare no conflict of interest. The funders had no role in the design of the study; in the collection, analyses, or interpretation of data; in the writing of the manuscript; or in the decision to publish the results.

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
