# Peer review of "The Material Matters: Sorption/Desorption Study of Selected Estrogens on Common Tubing or Sampling Materials Used in Water Sampling, Handling, Analysis or Treatment Technologies"

_applsci, doi:10.3390/app13053328_

Round 1

Reviewer 1 Report

The work reported by Odehnalova et al. and entitled: “The material matters: Sorption/desorption study of selected estrogens on common tubing or sampling materials used in water sampling, handling, analysis or treatment technologies” is not very novel or scientifically sound. Most of the work could be found in the comprehensive review of Schafer et al. Micropollutant sorption to membrane polymers: a review of mechanisms for Estrogens. DOI: 10.1016/j.cis.2010.09.006. The objective of this paper needs to be clearly written and elucidated prior to publication.

However, I believe this work could be published after carefully considering the following comments:

Abstract:

Line 11: µg/L not ug/L

Line 17: PVC, PTFE, PVDF and EPDM (abbreviations should be defined in the abstract)

Introduction:

Page 1, Line 32: Plastic and rubber materials for Technological equipment. “cite this recent reference: https://doi.org/10.3390/polym14193993 

Page 1, Line 36: The authors need to cite some reviews or articles on advanced AOP processes.

Page 2, Line 45: poorly written. It is hard to understand the sentence. Rewrite.

Lines 49-53: There are many other micro-pollutants that have more severe environmental impact. The authors follow only reports from European Union? What’s going on? The authors need to clearly explain the reason for using Estrogens in their study. Justification is needed to convince the readers.

Why did the authors only use LC-MS for characterizing their micro-pollutants and not other techniques?

The authors did not mention anything about the molar masses of their polymeric samples that were tested.

Author Response

Comment: Line 11: µg/L not ug/L

Corrections: correction made

Comment: Line 17: PVC, PTFE, PVDF and EPDM (abbreviations should be defined in the abstract)

Corrections: abbreviations were defined

Comment: Page 1, Line 32: Plastic and rubber materials for Technological equipment. “cite this recent reference: https://doi.org/10.3390/polym14193993

Corrections:

Comment: Page 1, Line 36: The authors need to cite some reviews or articles on advanced AOP processes

Corrections: New citations were added [1-3].

  1. Miklos DB, Remy C, Jekel M, Linden KG, Drewes JE, Hubner U. Evaluation of advanced oxidation processes for water and wastewater treatment - A critical review. Water Research. 2018;139:118-31. doi: 10.1016/j.watres.2018.03.042.
  2. Gogate PR, Patil PN. Combined treatment technology based on synergism between hydrodynamic cavitation and advanced oxidation processes. Ultrasonics Sonochemistry. 2015;25:60-9. doi: 10.1016/j.ultsonch.2014.08.016.
  3. Kanakaraju D, Glass BD, Oelgemoller M. Advanced oxidation process-mediated removal of pharmaceuticals from water: A review. Journal of Environmental Management. 2018;219:189-207. doi: 10.1016/j.jenvman.2018.04.103.

Comment: Page 2, Line 45: poorly written. It is hard to understand the sentence. Rewrite

Corrections: The sentence is rewritten: “Moreover, the sorption could be enhanced by the surface roughness and microporosity of the polymer material.”

Comment: Lines 49-53: There are many other micro-pollutants that have more severe environmental impact. The authors follow only reports from European Union? What’s going on? The authors need to clearly explain the reason for using Estrogens in their study. Justification is needed to convince the readers.

Answer: Estrogens were chosen for this study's focus for several reasons. First, Kow lipophilicity values indicate that they tend to adsorb on sediments or materials in the laboratory. Second, they were followed by the Europen watch list in the past as endocrine disruptors. According to studies focusing on this issue, the risk associated with estrogens is high. Since many micropollutants have a severe environmental impact, a system for calculating risk coefficients was developed to prioritise the most severe ones.

Corrections: The sentence was, and citation were added: “Due to their hydrophobicity values expressed as log Kow (see Table S1) and the high-risk coefficients, estrogens represent relevant model compounds [10, 11]

  1. Yang Y, Zhang XR, Jiang JY, Han JR, Li WX, Li XY, et al. Which Micropollutants in Water Environments Deserve More Attention Globally? Environmental Science & Technology. 2022;56(1):13-29. doi: 10.1021/acs.est.1c04250.
  2. Zhou SB, Di Paolo C, Wu X, Shao Y, Seiler TB, Hollert H. Optimization of screening-level risk assessment and priority selection of emerging pollutants - The case of pharmaceuticals in European surface waters. Environment International. 2019;128:1-10. doi: 10.1016/j.envint.2019.04.034.

Comment: Why did the authors only use LC-MS for characterizing their micro-pollutants and not other techniques?

Answer: The LC-MS technique was used because it is sensitive enough to determine very low concentrations corresponding to real concentrations of estrogens in the environment we wanted to mimic. – no changes in manuscript

Corrections: no correction was made

Comment: The authors did not mention anything about the molar masses of their polymeric samples that were tested.

Answer: Molar masses of tested materials were found as not relevant to be stated in this study. Molar masses of monomeric units can be easily found elsewhere, but the molar masses of final polymers can vary a little because of possible additives and heterogeneities. Nevertheless, the aim of our experiments was to assess the sorption/desorption of estrogens in contact with the material surface.

Reviewer 2 Report

The study aimed to evaluate the adsorption capacity of six polymers - commercially used in laboratory devices - to adsorb estrogen and evaluate the percentage of error resulting from the results of measuring the concentration of the estrogen in the ng/L concentration range. The study is a valuable addition in the field of micropollutant analysis, as it draws the attention of researchers in this field to consider the adsorption/absorption factor during analysis by the used plastic materials in the analysis devices or sampling tools. I recommend that the study be published in the Journal of Materials after addressing the following points:

-        In the estrogen’s adsorption test, the polymers were immersed in 2 liters of water containing the estrogens for 30 minutes. Why was 30 min chosen? Does estrogen adsorption reach equilibrium at 30 min? I suggested studying the adsorption kinetics and determining the required time to reach equilibrium.

-        Why was the adsorption time chosen as 12 hours with EPDM, while with other polymers it was 30 min?

-        Pieces of polymers with a surface area of 1000 cm2 were used to test adsorption. Are the masses of these polymers equal, and if they are unequal, is there an effect of mass on adsorption?

-        In estrogen extraction from samples was performed using SPE Oasis® HLB. Does this tool have the ability to adsorb estrogen as well?

-        Can you explain the nature of the interaction between estrogens and polymers and why the Tygon S3™ material is higher in adsorption-desorption?

-        Figures must be placed after their respective text.

-        In line 281, “Estrogen cumulative concentrations desorbed from the EPDM surface are shown in Fig. 2C.” Please correct the Figure number.

Author Response

Comment: In the estrogen’s adsorption test, the polymers were immersed in 2 liters of water containing the estrogens for 30 minutes. Why was 30 min chosen? Does estrogen adsorption reach equilibrium at 30 min? I suggested studying the adsorption kinetics and determining the required time to reach equilibrium.

Answer: Current research in the field of removing micropollutants with new technologies aims to reduce contact time and improve the possibility of implementation in practice. We realized a set of the preliminary tests, from which the 30mins resulted as a relevant time. That's why we chose 30 minutes. They are long enough time for sample handling and relevant experimental designs.

Comment: Why was the adsorption time chosen as 12 hours with EPDM, while with other polymers it was 30 min?

Answer: In the case of EPDM as the most suitable material according to the screening test, the experiment was extended due to the need for more information on sorption to this polymer which is not available in the current literature. At the same time, the possibility of use in longer-term experiments was verified, and the suitability of EPDM for these cases was confirmed.

Comment: Pieces of polymers with a surface area of 1000 cm2 were used to test adsorption. Are the masses of these polymers equal, and if they are unequal, is there an effect of mass on adsorption?

Answer: The masses of polymers were not equal, but they were not relevant to the aim of this study. The polymer masses play a role in the case of, e.g. sorption on microplastics where the surface area is not possible to measure. In our case, we want to evaluate the losses made by passing through the hose or in contact with other laboratory material; thus, only the surface area is a relevant parameter.

 Comment: In estrogen extraction from samples was performed using SPE Oasis® HLB. Does this tool have the ability to adsorb estrogen as well?

Answer: When working with environmentally relevant concentrations, the samples needed to be concentrated and based on a literature search, Oasis® HLB SPE columns were selected for this purpose. These columns are made of polypropylene. According to the available literature, estrogens are only minimally sorbed to PP. At the same time, the estrogens were quantified using matrix-matched calibration, where the calibration solutions undergo the same procedure as samples, to avoid the influence of this factor on the results.

Comment: Can you explain the nature of the interaction between estrogens and polymers and why the Tygon S3™ material is higher in adsorption-desorption?

Answer: This topic is discussed in lines 214-238. The exact mechanism remains still unknown, but we assumed that hydrophobic interactions play a main role, except for Tygon. In this case, the interaction is very weak. And based on desorption experiments, it is more likely to be related to the diffusion of estrogen molecules into the material's pores in exchange for a plasticiser, for example.

Comment: Figures must be placed after their respective text.

Corrections: Figures were moved to the proper places.

Comment: In line 281, “Estrogen cumulative concentrations desorbed from the EPDM surface are shown in Fig. 2C.” Please correct the Figure number.

Corrections: The number of Figure was corrected: “Fig. 3C”

Round 2

Reviewer 1 Report

I believe the authors have satisfactorily answered my comments.